# Conformational Dynamics of Lipoxygenases and Their Interaction with Biological Membranes

**DOI:** 10.3390/ijms25042241

**Published:** 2024-02-13

**Authors:** Fulvio Erba, Giampiero Mei, Velia Minicozzi, Annalaura Sabatucci, Almerinda Di Venere, Mauro Maccarrone

**Affiliations:** 1Department of Clinical Science and Translational Medicine, Tor Vergata University of Rome, Via Montpellier 1, 00133 Rome, Italy; erba@uniroma2.it; 2Department of Experimental Medicine, Tor Vergata University of Rome, Via Montpellier 1, 00133 Rome, Italy; mei@med.uniroma2.it; 3Department of Physics and INFN, Tor Vergata University of Rome, Via Della Ricerca Scientifica 1, 00133 Rome, Italy; velia.minicozzi@roma2.infn.it; 4Department of Biosciences and Technology for Food Agriculture and Environment, University of Teramo, Via Renato Balzarini 1, 64100 Teramo, Italy; alsabatucci@unite.it; 5Department of Biotechnological and Applied Clinical Sciences, University of L’Aquila, Via Vetoio, Coppito, 67100 L’Aquila, Italy; 6European Center for Brain Research (CERC), Santa Lucia Foundation IRCCS, 00143 Rome, Italy

**Keywords:** lipoxygenase, membrane binding, conformational flexibility, interdomain interaction, molecular dynamics

## Abstract

Lipoxygenases (LOXs) are a family of enzymes that includes different fatty acid oxygenases with a common tridimensional structure. The main functions of LOXs are the production of signaling compounds and the structural modifications of biological membranes. These features of LOXs, their widespread presence in all living organisms, and their involvement in human diseases have attracted the attention of the scientific community over the last decades, leading to several studies mainly focused on understanding their catalytic mechanism and designing effective inhibitors. The aim of this review is to discuss the state-of-the-art of a different, much less explored aspect of LOXs, that is, their interaction with lipid bilayers. To this end, the general architecture of six relevant LOXs (namely human 5-, 12-, and 15-LOX, rabbit 12/15-LOX, coral 8-LOX, and soybean 15-LOX), with different specificity towards the fatty acid substrates, is analyzed through the available crystallographic models. Then, their putative interface with a model membrane is examined in the frame of the conformational flexibility of LOXs, that is due to their peculiar tertiary structure. Finally, the possible future developments that emerge from the available data are discussed.

## 1. Introduction

Lipoxygenases (LOXs) are dioxygenases that play a key role in the metabolism of polyunsaturated fatty acids (PUFAs)—among which is arachidonic (eicosatetraenoic) acid—in a large variety of living cells [1] ranging from microorganisms [2] to plants [3] and mammals [4]. Such a ubiquitous distribution is suggestive of relevant biological roles conferred to LOXs by natural evolution [5,6]. The reaction catalyzed by LOXs is the oxygenation of PUFAs, i.e., the insertion of molecular oxygen (O_2_) in their acyl chain with the generation of hydroperoxyl (HOO-) groups at different positions. Insertion of O_2_ is specific for each LOX isoform that is indeed named with the number of the carbon atom where O_2_ has been bound. LOXs products initiate crucial biosynthetic pathways in living organisms. In mammals, 5- and 15-LOXs induce the synthesis of important signaling molecules [1,7], such as leukotrienes (5-LOX) and lipoxins (15-LOX), that play a crucial role in inflammation and immunity [8]. They are also involved in the development of pathological atherosclerosis [9], as a direct consequence of their ability to bind low density lipoproteins [10]. Plant lipoxygenases (8-, 13-LOXs) are instead involved in germination and growth, as well as in pathogen resistance [3]. Additionally, some LOXs can modify the structure of lipid bilayers to reach specific metabolic goals: for example, in plants, oxidation of membranes by LOXs drives leaf senescence or lipid mobilization during the germination phase [3]; in animals, 15-LOX of reticulocytes attack the mitochondria envelope and promote the maturation of red [11,12]. 

The first biochemical characterization of a (plant) LOX dates back to the 1970s [13,14] but, despite the widespread presence of LOXs in the seeds of legumes [15], the first tridimensional structure of a soybean 15-LOX (also known as LOX-1) was obtained only 20 years later [16,17]. The main 3D features of soybean 15-LOX are: (i) the existence of two rather distinct structural domains, and (ii) the presence of a non-heme catalytic iron located in the C-terminal domain of the polypeptide chain [16,17]. For several years, only soybean 15-LOX 3D structure was available, and thus was used as a template to model human 5-, 12-, and 15-LOXs [18]. A more reliable model for mammalian (and in particular human) LOXs was possible when rabbit reticulocyte 15-LOX was crystallized [19] and its preliminary structure was determined [20] and then refined [21]. The further characterization of human [22,23], porcine [24], and coral [25] LOXs has provided evidence that a common 3D architecture does exist in animal LOXs. 

Starting in the 1970s, the attention on LOXs has increased considerably, as it appears from the number of papers published since then (Figure 1A). Human 5-LOX and 15-LOX are clearly the most studied members (Figure 1A) due to their major impact on health and disease conditions [26,27,28,29,30]. Of note, only a small number of studies have interrogated the interaction of LOXs with bio-membranes (Figure 1B), despite two fundamental questions arising from this event: (i) How do soluble enzymes (such as LOXs) search and find their substrates in a rather peculiar environment like lipid bilayers? (ii) How does the interaction with bio-membranes modulate the activity of LOXs? The first issue is obviously not specific for LOXs, yet these enzymes may represent a paradigmatic example to shed light on other membrane-interacting proteins [31].

The second issue seems to be of great relevance for LOXs, especially for human 5-LOX, whose activity leads to the synthesis of bioactive compounds—leukotrienes—from arachidonic acid [26,29]. Human 5-LOX is, indeed, the only member of the LOXs family that is present both in the cytoplasm and in the nucleus of a cell and that is able to bind different types of membranes (plasma membranes, nuclear membranes) both directly or through the specific 5-LOX activating protein, FLAP [27].

Here, we summarize the main structural characteristics of LOXs, in particular their inter-domain interactions and dynamic properties in the frame of the enhanced flexibility that is peculiar of this class of oxidoreductases. Then, we examine the typical features of the N-terminal β-sandwich domain, to discuss its mechanistic role in membrane binding and to compare the putative position that different LOX isoforms might assume in lipid bilayers. Finally, we discuss possible future developments of the research on LOXs-membrane interaction based on both experimental and theoretical approaches. 

## 2. Insights into the Architecture of LOXs

Despite the huge amount of data available on phylogenetics, biological activity, design and action of inhibitors, and in vivo localization of LOXs, these proteins remain quite elusive from the structural point of view. In fact, a limited number of 3D structures are as yet available because of crystallization problems due to unstable segments in the protein sequence. The first two characterized LOXs are a plant enzyme, namely soybean 15-LOX (the first ever to be crystallized), and its mammalian counterpart, 12/15-LOX from rabbit reticulocyte. These two proteins share a limited sequence homology (<24%, Figure 2) and have different molecular weights (94,000 and 77,000, respectively); nonetheless, they display the same 3D organization. In particular, X-ray measurements [16,20] revealed the presence of two different domains, namely an N-terminal β-barrel PLAT (Polycystin-1, Lipoxygenase, Alpha-Toxin) domain, and a larger C-terminal domain that is mainly characterized by α-helices (Figure 3 and Figure 4). The two domains are connected by a short flexible linker and play different functional roles: the C-terminal contains the catalytic site [16,20], while the N-terminal has regulatory functions and, for instance, influences the membrane binding ability of mammalian enzymes [32]. This general structural organization is also highly conserved in other human variants, such as 5-, 12-, and 15-LOX [22,23,33], and in coral 8-LOX [25], despite the degree of sequence identity with both plant and rabbit LOXs being on the average quite low (Figure 2). 

The composition of the two N- and C-terminal domains is significantly different in all the above-mentioned LOX structures. The smaller β-barrel N-terminal is rather dense, mainly containing tightly packed hydrophobic residues; the C-terminal core is instead characterized by large cavities suitable for oxygen transit and substrate [35,36,37,38]. The two domains are separated by a large gap, yet they are not fully independent. In Figure 3 and Figure 4, the analysis of the domain–domain interface is reported in terms of “contact maps”, which indicate the closest points of contact between the two protein sections. Plant and mammal lipoxygenases share a similar architecture, in which a few (≈4) main groups of contacts characterize the interaction between the two domains (Figure 3 and Figure 4). However, the positions of such contacts along the polypeptide chain of soybean 15-LOX are not the same reported for mammalian enzymes or coral 8-LOX, due to the larger size of the plant enzyme. Indeed, except for soybean 15-LOX, the other 5 enzymes considered in this study exhibit the most relevant contacts in similar locations, namely at positions 15, 25, 60, and 100 in the N-terminal domain, and approximately at positions 160, 390, and 620 in the C-terminal domain (Figure 3 and Figure 4). A more detailed analysis of these regions reveals a high homology score among the five LOX polypeptide chains (Figure 5), suggesting that such amino acids might play an important role in the protein dynamics. In particular, it could be speculated that these regions communicate movements of the N-terminal to the C-terminal and vice versa, in analogy to the “hot spots” found in the contact networks of oligomeric protein inter-subunit surfaces [39]. Furthermore, it should be stressed that many of the conserved or semi-conserved residues are histidines and aromatic amino acids (Figure 5), whose large side chain fills the gap at the domain–domain interface.

## 3. Structural Flexibility of LOXs

One important feature of a protein structure is the intrinsic plasticity due to the presence of inner empty cavities, water molecules, and inter-domains movements. LOXs accomplish, at the same time, quite different tasks, hosting the acyl chain of polyunsaturated fatty acids and binding membranes, two functions that require a certain degree of elasticity. Small angle X-ray scattering experiments suggested that rabbit 12/15-LOX [43] and human platelet 12-LOX [44] display a certain degree of conformational flexibility due to both local and global structural changes. Temperature- and pressure-dependent dynamic fluorescence data [45,46] led to a similar conclusion. Local flexibility was probed by the conformational changes induced by an eicosatetraynoic acid (ETYA) inhibitor, and was thus associated with the active site in the C-terminal domain [46]. These data are compatible with the mobility of a few α-helix segments that characterize the opening of the catalytic pocket in rabbit 12/15-LOX [21]. Instead, global flexibility arises from interdomain movements [43,44], as also suggested by molecular dynamics simulations [47]. The few key contacts that characterize the domain–domain interface (Figure 3 and Figure 4) play a major role in regulating protein plasticity, especially if they are characterized by a specific aromatic side chain. For instance, Y98 is a highly conserved residue (Figure 5) that, in rabbit 12/15-LOX and in human 12-LOX, occupies a relevant position at the domain–domain interface (Figure 3B and Figure 4B). Its substitution with smaller amino acids (for instance phenylalanine or alanine) does not influence protein secondary and tertiary structures, but has a strong impact on enzyme catalysis and stability by modulating domain association and substrate binding [48]. Flexibility is fundamental for interdomain communication also in human 15-LOX [49] and coral 11-LOX [50]. In fact, it was proposed that the N-terminal domain could exert an allosteric regulation of LOX catalytic activity through residues at the domain-domain interface [50]. A highly conserved tryptophan in the FPCYRW segment (Figure 5) seems to be the best candidate to accomplish such a task, due to its strong interaction with the group of amino acids located in the C-terminal domain between positions 160 and 170 (Figure 3 and Figure 4). It should be noted that, within the same region, a lysine and a phenylalanine—a tyrosine in human 15-LOX—are always present in animal isoforms (Figure 5). Therefore, it is tempting to speculate that the proposed communication mechanism between the two protein domains [50] could be a common feature of all animal LOXs. Finally, the comparison between soybean and rabbit LOXs demonstrated that a higher flexibility of the mammal enzyme facilitates its membrane binding in both in vitro and ex vivo measurements [46], indicating that the global flexibility of (some) LOXs can directly modulate their interaction with lipid bilayers. 

## 4. Membrane Binding Ability of LOXs

### 4.1. Peculiar Features of the N-Terminal β-Sandwich Domain

The N-terminal domain of LOXs seems to perform several activities connected (both directly and indirectly) with membrane binding. For instance, site-directed mutagenesis studies provided evidence that it is involved in the Ca^2+^-dependence of enzymatic activity [51]. In human 5-LOX, it plays a crucial role in the protein translocation to the nuclear envelope [52], while in other mammalian LOX isoforms, it functions as a main regulator of different protein activities, including recognition and binding to lipid bilayers [32,43]. A possible direct involvement of the β-sandwich domain in membrane binding has been suggested since the 1990s. In fact, when the first structures of a plant (1993) and a mammalian (1997) enzyme were solved, a similarity with the colipase binding domain of pancreatic lipase and with the so-called C2-domain of several lipid-binding proteins was discovered [20,53,54]. Engineered 15-LOX, lacking the β-sandwich domain, showed a reduced membrane-binding ability [55,56]. Conversely, enzymatic cleavage of soybean 15-LOX produced a trimmed enzyme—called “mini-15-LOX”—with enhanced membrane-binding ability but very low structural stability [57,58]. The peculiar structural features of the β-sandwich domain of the six LOX isoforms studied here are shown in Figure 6 and Figure 7. The core of the domain is a compact ≈ 25 Å × 40 Å cylinder that obeys a rigid body dynamic. As already mentioned, the region facing the C-terminal domain contains several aromatic side chains, whose position is indicated in Figure 6 and Figure 7. Very flexible loops characterize the two extremities of the β-sandwich. These loops also contain a relevant number of aromatic amino acids (Figure 5 and Figure 6). Mutations of some of these residues (e.g., W13, W75, and W102) in human 5-LOX considerably reduced the protein interaction with synthetic vesicles of different composition, proving that their side chains play a relevant role in membrane binding [59]. Indeed, an accurate characterization of the protein–membrane surface through fluorescence [Förster] resonance energy transfer (FRET) measurements has demonstrated that the ring of W75 in human 5-LOX penetrates the bilayer below the polar heads of the phospholipids [60].

Thanks to the high flexibility of the β-sandwich loops, such a deep insertion into the hydrophobic core of the membrane would help further protein anchoring to the membrane, thus facilitating the insertion of other aromatic residues present on the catalytic C-terminal domain [60]. Large amino acid side chains (namely H53 and F69) are also present in the mobile loops of both soybean 15-LOX and rabbit 12/15-LOX, supporting that such contacts are indeed important for both membrane recognition [31] and substrate acquisition [61]. Very recently, Garcia and co-workers [62] have demonstrated that also the N-terminal loops of coral 8-LOX and human 15-LOX are fundamental for membrane binding. In line with this, deletion of a few amino acids in these regions impairs the membrane binding activity of both enzymes [62]. Finally, truncation of the whole N-terminal β-sandwich in human 12-LOX reduces the protein–membrane interaction by one-half [63], proving again the relevance of this domain for the lipid-binding mechanism of LOXs.

### 4.2. Membrane-Binding Modeling

Despite the fact that the presence of the N-terminal barrel is undoubtedly relevant for membrane binding by LOXs, studies on human variants have suggested that the protein surface involved in such an interaction must also include a large portion of the C-terminal catalytic domain. For instance, in 2005, Tatulian and co-workers proposed a pioneering model of human 5-LOX where binding to a membrane would involve C-terminal residues (like lysine 183, phenylalanine 197, and tryptophans 201 and 599), most of which are characterized by aromatic side chains [60]. A similar result was also obtained by site directed mutagenesis of rabbit 12/15-LOX. In this case, the substitution of tryptophan 181 and leucine 195 with aspartic acid yielded a considerable decrease in membrane binding ability [61]. Models of soybean, rabbit, coral, and human LOXs bound to a fluid POPC (1-palmitoy-2-oleyl-*sn*-glycero-3-phosphocholine) membrane are reported in Figure 8. All the mammal enzymes exhibited a similar angular orientation (≈45°) of the N-terminal β-sandwich with respect to the plane of the bilayer (Figure 8B,D–F). Such a configuration was already proposed for the mammalian 15-LOX and human 5-LOX in previous studies [54,60]. Instead, the 15-LOX plant enzyme has a different orientation, with the N-terminal domain being twisted 90° towards the membrane plane (Figure 8A). Despite such a configuration would suggest a different anchoring to the membrane; also, in this case, the C-terminal contribution to the protein–lipid contact interface was not negligible (Figure 4A). This observation is consistent with at least two experimental findings: (i) the N-terminal depleted protein (soybean mini-15-LOX) has an enhanced membrane binding capacity [57], that is diagnostic of both a good affinity of the C-terminal domain alone for lipid bilayers, and a regulatory role of the N-terminal domain also in intact soybean 15-LOX; (ii) as demonstrated by FRET measurements [64], the plant isoform preferentially binds to bilayers containing a selected group of bioactive lipids called “endocannabinoids” [65] which must directly interact with the protein C-terminal domain in order to be oxidized by its enzymatic activity.

### 4.3. Measure of LOXs Binding to Synthetic Vesicles

In the last twenty years, several experimental data on both kinetic- and equilibrium-binding assays of LOXs to model membranes have been published. The main results of these studies are summarized in Table 1 for different LOX isoforms, together with the used experimental conditions, methodologies, and type of synthetic membranes. The relevant information contained in Table 1 concerns the low values of the dissociation constant (K_d_) obtained in the case of fluid membranes (0.2 < K_d_ < 0.7 μM). This observation and the flexible loops involved in the interaction—shown in Figure 6 and Figure 7—would suggest that mobility at the protein–membrane interface is a prerequisite necessary for binding. In line with this, a recent study on human 5-LOX conformational states has confirmed that this is the case, demonstrating that the residues laying at the protein–membrane interface and facing the entrance of the C-terminal cavity are the most mobile and show the highest B-factor values [30]. Although this property is yet to be demonstrated for other LOX isozymes, the values reported in Table 1 indicate that the propensity to bind fluid membranes seems a common feature to all LOXs studied so far. Another important issue is the presence of Ca^2+^ ions (in the 0.2–5.0 mM range) in all analyses shown in Table 1. The importance of Ca^2+^ in promoting human 5-LOX [51,52], mammalian 15-LOX [49,61], coral 8-LOX [25], and soybean 15-LOX [64] interaction with membranes is well known; yet, its contribution to membrane binding by LOXs clearly depends on experimental conditions [28]. For instance, activation of 5-LOX can be achieved in vivo in a Ca^2+^-independent manner, if cells are under intense stress [68]. Also, in vitro binding of 5-LOX to nanodiscs was obtained in the absence of Ca^2+^, provided that both the arachidonic acid substrate and FLAP (5-LOX-activating protein) were present [69]. Another point of concern is how Ca^2+^ ions can support binding at the β-barrel domain of LOXs. Measurements performed on the sole N-terminal section of human 5-LOX demonstrate that: (i) this domain alone displays a much higher membrane-binding ability than that of all the whole protein isoforms (Table 1); and (ii) its interaction with the bilayer is strongly affected by the presence of Ca^2+^ ions (Table 1). Most of the Ca^2+^ binding sites identified in mammal LOXs reside in their N-terminal domain [51,59], whose truncation impairs the membrane-binding process [32]. However, the main driving force of LOX–membrane interaction has been ascribed to the presence of aromatic [59] and hydrophobic [61] side chains, while the role of Ca^2+^ ions should be restricted to a further stabilization of such an interaction through bridging negatively charged amino acids to phospholipid heads [61]. Of note, Ca^2+^ binding to LOXs is not restricted to the N-terminal domain, and in fact, N-terminal-truncated mammal enzymes do not completely lose their capacity to bind membranes in the presence of Ca^2+^ [61]. Consistently, the binding ability of the C-terminal catalytic domain of LOXs is also conserved in some bacterial isozymes that lack the N-terminal section [70]. In other prokaryotic LOXs, the N-terminal β-sandwich is not essential for membrane binding, but anchoring to lipid bilayers is provided by an extra α-helix that does not belong to the catalytic domain [71]. Against this background, it is apparent that the dependence of protein–lipid interaction on Ca^2+^ ions remains to be clarified.

### 4.4. Lessons Learned from Prokaryotic LOXs

Genome analysis demonstrated that LOXs are not widely expressed in bacteria (Horn et al., 2015). As a consequence, understanding the biology of bacterial LOXs is still underdeveloped compared to that of eukaryotic enzymes. Remarkably, the first crystallized prokaryotic LOX from *Pseudomonas aeruginosa* has revealed the absence of the N-terminal domain which is typical of animal and plant isozymes [70]. Instead of it, a cluster of additional α-helices was found, and was shown to be characterized by a high flexibility [73] that made the enzyme more soluble [74]. Despite the absence of the N-terminal β-sandwich domain, *Pseudomonas* LOX efficiently binds to membranes, thus being particularly effective in the hemolysis of human red cells [75]. Conversely, LOX of the cyanobacteria *Cyanothece* sp. contains a few β-strands (which resemble the eukaryotic β-sandwich structure), plus an N-terminal α-helix, similar to those of *Pseudomomas aeruginosa* [71]. In this case, the β-strands domain is not important for membrane binding, but it is for catalytic activity, and removal of the N-terminal α-helix impairs enzyme interaction with liposomes [71]. As mentioned above, in the past, both gene truncation and enzymatic cleavage have been used to separate the catalytic domain from the N-terminal β-barrel of eukaryotic LOXs. In both cases, opposite effects in plant and mammalian enzymes have been obtained. In soybean 15-LOX, removal of the N-terminal domain yielded a less stable fragment (mini-15-LOX), characterized by increased binding to fluid lipid bilayers [57,58]. In the case of mammalian 15-LOX and 12-LOX, C-terminal mini-LOXs were also produced and showed a worse interaction with membranes [55,63]. Instead, in a fungal LOX, the deletion of the N-terminal domain led to an inactive mini-LOX form [76]. One possible explanation for these differences may reside in the low flexibility of the full-length soybean 15-LOX [45,46,77] that would instead acquire a more elastic structural conformation upon N-terminal deletion [32]. In this context, it seems noteworthy that evolution has conserved the overall structural properties of the catalytic domain of LOXs better than that of the peripheral structural motifs, both in eukaryotic and prokaryotic organisms [2,6]. Taken together, the available data suggest that alternative structural strategies may exist (e.g., peripheral α-helices, β-strands, or more complex β-sandwich domains) in different species to drive membrane binding by LOXs, and hence to regulate the biological function of these key oxygenases [71].

### 4.5. Oligomerization: An Additional Determinant of Membrane Binding by LOXs

A relevant feature of LOXs is their propensity to form oligomers. Since their discovery in the 1970s and structural characterization at the beginning of the 1990s, eukaryotic LOXs have been assumed to act as monomers. Then, in 2011, two groups independently demonstrated that both rabbit 12/15-LOX [44] and human 5-LOX [78] may form dimers. Dimerization of the rabbit isozyme depends on ligand binding, and probably involves hydrophobic contacts between the two α-helices of each monomer [48]. Such a non-covalent interaction has been proposed to give rise to the allosteric regulation driven by a 15-LOX product, namely 13-(*S*)-hydroxyoctadecadienoic acid [79,80]. In the case of 5-LOX, the presence of several exposed cysteines, the diamide-induced oligomerization, and a protein mutant lacking four cysteine residues suggested a specific monomer–monomer orientation in dimer assembly [78]. In particular, according to docking minimization, it was proposed that monomers couple in a head-to-tail configuration [78], and that four cysteine residues (at positions 159, 416, 418, and 300) are involved in such an interaction [78]. The position of these residues in human 5-LOX is shown in Figure 9 from different angles, together with the amino acids tryptophan 75 and histidine 195, that presumably penetrate more deeply into the membrane, as shown in Figure 8D. 

A relatively small angle between the alignment of the cysteines and that of the membrane-interacting residues occurs, indicating that the cysteines involved in monomer-monomer interaction are close to the protein-lipid interface. Such a configuration suggests that the presence of a second monomer might impair membrane binding, and that dimerization might be a strong regulator not only of the enzyme catalytic activity, but also of other LOX functions, indirectly connected to its interaction with lipid bilayers [78]. In line with this, recently, the existence of different concomitant oligomeric states (dimers, tetramers, and hexamers) of human 12-LOX has been also reported [33]. Thanks to high resolution cryo-electron microscopy, a hierarchic association mechanism has been discovered, according to which monomers associate into dimers that only subsequently aggregate into dimers of dimers or trimers of dimers [33]. Interestingly, the structure of the larger oligomers (tetramers and hexamers) displays a reduced membrane-binding surface, similarly to what is observed in dimeric 5-LOX (Figure 9). Moreover, the active site entrance of such multimeric 12-LOX is hidden by the association of a couple of dimers, reducing enzyme activity overall [33]. These results have suggested the hypothesis that oligomerization may help to regulate the enzyme biological function, providing, at the same time, “storage pools” of LOXs within the cell [33]. 

## 5. Future Perspectives

Pockets, cavities, and tunnels are probably the most striking structural features of proteins and enzymes. Far from being rigid scaffolds, such molecular architectures are dynamic entities that arise from the flexible 3D folding of polypeptide chains [81]. In LOXs, these cavities play a crucial role by connecting the buried, iron-containing active site to the protein surface, so that their dynamics directly governs substrate acquisition and product release [1]. In this context, the interface between the two domains is expected to play a relevant role in coordinating membrane binding (driven by the N-terminal domain) and enzymatic activity (performed by the C-terminal domain). The hypothesis that interdomain motions might influence adaptation of the substrate to the binding site was recently supported by studies on human 5-LOX inhibition [35]. It has been shown that a natural compound such as 3-acetyl-11-keto-beta-boswellic acid (AKBA) fits in the pocket between the N- and C-terminal domains, impairing their reciprocal movements and changing the enzymatic regiospecificity [82]. It is worth mentioning that iron and the binding site of AKBA are separated by ≈3 nm, and that an allosteric mechanism characterizes the action of this inhibitor [35]. This observation seems important, because it opens new perspectives in the pharmaceutical research of unconventional drugs that could regulate an active site from a distance. Besides AKBA, the N-terminal/C-terminal interface is also the target of other molecules, such as the coactosin-like protein (CLP) that can stabilize 5-LOX and modulate its Ca^2+^-dependent activities, including membrane binding [83]. In particular, CLP and membranes act in synergy during the 5-LOX-dependent synthesis of leukotrienes, and a role for CLP as a chaperone in a ternary complex (CLP/5-LOX/membrane) was proposed [84], suggesting that in vivo, the interaction of 5-LOX with lipid bilayers might be also influenced by the action of inter-domain binding compounds. Previous in silico simulations of mammalian 5- and 15-LOX have demonstrated that the angle between the two domains may have a variance of ≈7–10 degrees during the simulation time [47,85]. Whether such a change also occurs when the protein binds a lipid bilayer remains an open question. Molecular dynamics in the presence of membranes containing different types of lipids might give new insights on the protein dynamics upon binding. A parallel experimental challenge is the design, construction, and characterization of single-point mutants in which specific key residues (such as those indicated in Figure 3 and Figure 4) are changed to interrogate which consequences might occur in the protein conformational dynamics, activity, and membrane-binding properties. The substitution of amino acids involved in membrane anchoring is another possible strategy to unravel the complex mechanism of the action of LOXs and the differences among prokaryotic, plant, and mammalian isoforms.

## 6. Conclusions

The membrane binding ability of LOXs has been documented since 1975 [86]. The ensemble of lipid constructs involved in LOXs interaction includes liposomes, lipid bodies, lipoproteins, reticulocytes, mitochondria, and, in the case of human 5-LOX, also the nuclear membrane via FLAP [10,87,88,89,90]. The results reported so far indicate that the interaction of LOXs with membranes is a complex process which requires high protein flexibility. In some LOX isozymes, such a flexibility is provided by a certain mobility of the N-terminal domain, which might play a role in the indirect regulation of conformational changes occurring at a distance, i.e., in the substrate binding cavity of the C-terminal domain. This observation is suggestive of a new role for the domain-domain interface of LOXs that can host in its large and flexible cavity a new class of enzyme inhibitors, as indeed shown very recently [82]. Furthermore, oligomerization of mammalian LOXs has revealed new and important roles for the C-terminal domain, probably not only limited to catalytic activity, but also engaged in membrane binding.

The variety of membrane targets of LOXs is accompanied by important products of enzyme activity that play fundamental roles in human life. For instance, human 5-LOX derivatives are involved in several diseases [4,91]. Thus, the development of inhibitors that can modulate its activity has become a priority in the pharmacological research [27,28,92]. LOX activity in plants is also important for humans because plant volatiles and jasmonic acids are enzyme products that can protect against pathogens [93,94]. These compounds are generally involved in growth regulation and plant defense mechanisms, so their serial production could be used in the food industry to develop new agents for pest control and food maintenance [7,95]. All these aspects make the regulation of LOX activity and membrane binding a central point for the next generation of scientists interested in the structure-to-function relationship of interfacial enzymes that are able to find hydrophobic substrates within the membranes and release hydrophilic products in the cell cytosol.

## Figures and Tables

**Figure 1 ijms-25-02241-f001:**
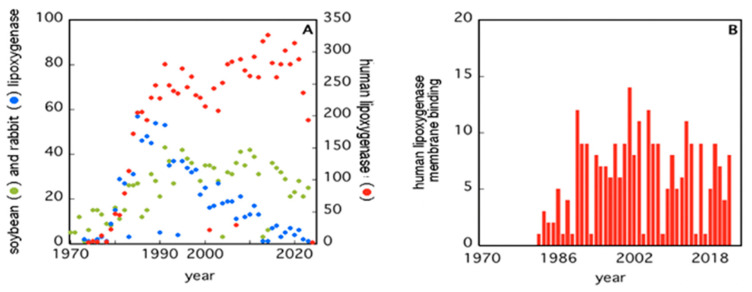
Number of papers retrieved from PubMed when using the keywords “soybean”, “rabbit”, “human”, and “lipoxygenases” (Panel (**A**)), or “human lipoxygenases membrane binding” (Panel (**B**)).

**Figure 2 ijms-25-02241-f002:**
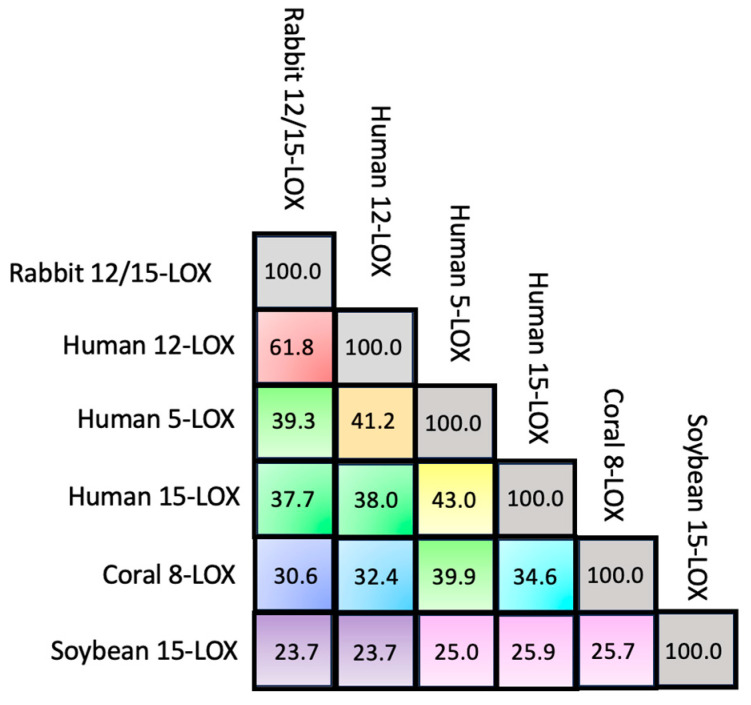
Percentage of sequence identity among the six LOX isoforms considered in this study. Sequence alignment and percentage identity have been obtained using the Align procedure available at www.uniprot.org/align accessed on 25 January 2024 [34].

**Figure 3 ijms-25-02241-f003:**
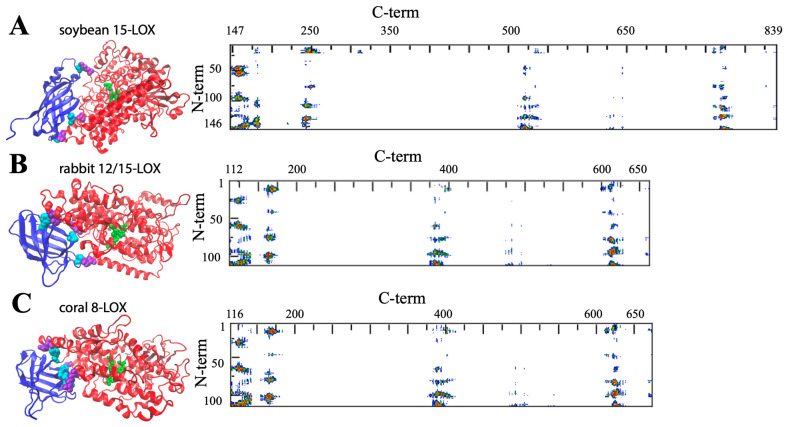
General structure of soybean 15-LOX (**A**), rabbit 12/15-LOX (**B**), and coral 8-LOX (**C**). The models are shown in a secondary structure rendering. Fe ligands are in green. Some representative contact residues between the two domains are shown in cyan and violet for the N- and C-terminals, respectively (A: V22, G54, F144, P157, K252, V520; B: S13, F62, Y98, H128, E169, Y614; C: P102, W106, F114, Q132, R167, E647). On the right side, the contact maps of the two domains are reported (red, yellow, green, and blue spots correspond to 7, 10, 13, and 16 Å inter-residues distances). The models reproduced in the figure have been obtained from the available PDB files (1f8n, 2p0m, and 2fnq, respectively), and the corresponding contact maps have been produced using the COCOMAPS software www.molnac.unisa.it/BioTools/cocomaps/ accessed on 20 January 2024 [40].

**Figure 4 ijms-25-02241-f004:**
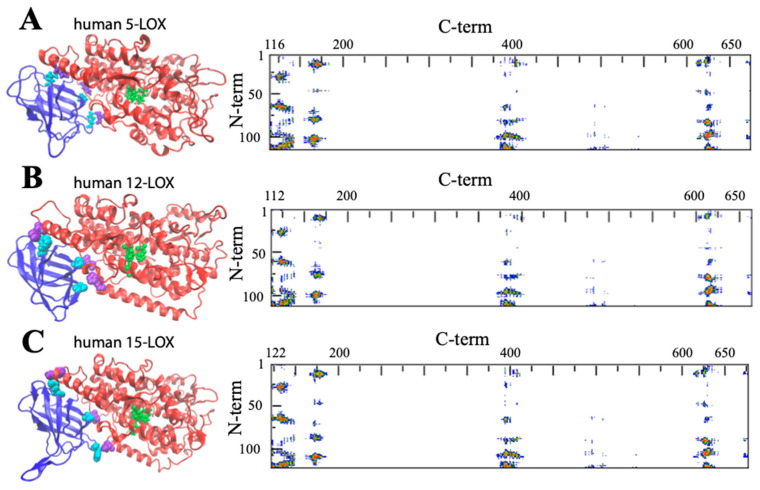
General structure of human 5-LOX (**A**), human12-LOX (**B**), and human 15-LOX (**C**). Graphic rendering and contact map color codes are the same used in Figure 3. Representative contact residues between the two domains are shown in cyan and violet for the N- and C-terminals, respectively (A: Q13, L67, Y101, H131, D171, H625; B: A12, F62, Y98, H128, E168, Y614; C: F14, L65, Y107, H131, T178, D625). The models reproduced in A, B, and C have been obtained from the available PDB files: (3o8y, 8ghb, and 4nre, respectively). Missing atoms in the 5-LOX structure were added using the Chimera interface to Modeller [41,42].

**Figure 5 ijms-25-02241-f005:**
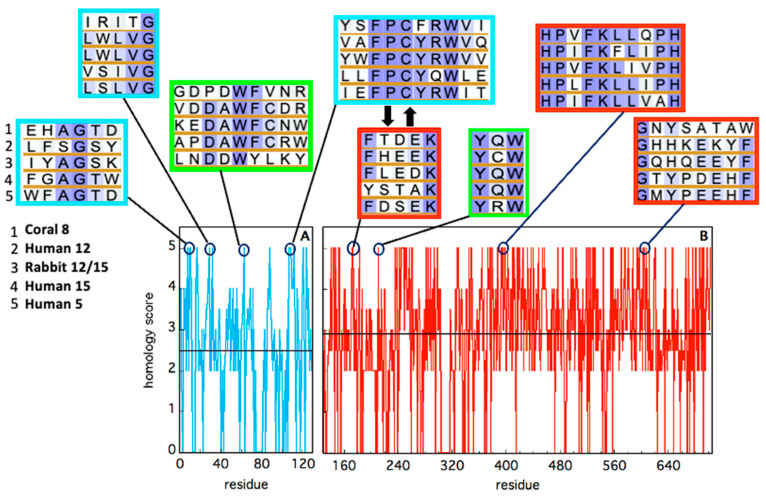
Sequence homology of the five animal LOX isoforms considered in this study, in the N-terminal (cyan) and C-terminal (red) domains. Alignment has been obtained using BLAST. Identity has been quantified by assigning an “homology score” using the following scheme: XXXXX = 5; XXXXZ = 4; XXXZZ = 3.5; XXXZB = 3; XXZZB = 2.5; XXZBO = 2.0; XZBOU = 0. The average score in each domain is reported as a black horizontal line. The local sequence segments containing residues involved in relevant domain-domain contacts (i.e., the red spots in Figure 3 and Figure 4) are reported in the cyan (N-terminal) and red (C-terminal) rectangular boxes. The two thick black arrows indicate two interacting segments of particular relevance, as discussed in the text. Putative conserved residues involved in protein–membrane interaction are shown in the two green rectangles.

**Figure 6 ijms-25-02241-f006:**
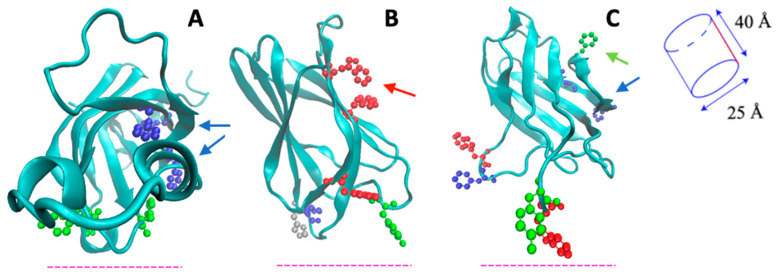
The three images represent the structures of soybean 15-LOX (**A**), rabbit 12/15-LOX (**B**), and coral 8-LOX (**C**) N-terminal domains. The dashed line (pink) represents the relative position of the membrane. The aromatic amino acids and histidine probably involved in protein–membrane interaction are shown in blue (PHE), green (TYR), red (TRP), and grey (HIS). The aromatic residues lying at the interface between the N- and C-terminal domains are indicated by the arrows in the corresponding color.

**Figure 7 ijms-25-02241-f007:**
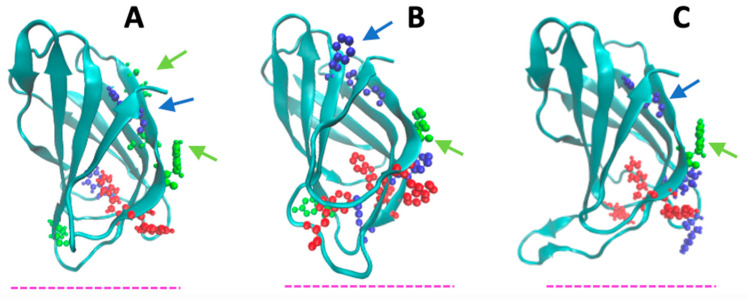
N-terminal domain images of human 5- (**A**), 12- (**B**), and 15-LOX (**C**). The dashed line (pink) represents the relative position of the membrane. The aromatic amino acids and histidine probably involved in protein–membrane interaction are shown in blue (PHE), green (TYR), red (TRP), and grey (HIS). The aromatic residues lying at the interface between the N- and C-terminal domains are indicated by the arrows in the corresponding color.

**Figure 8 ijms-25-02241-f008:**
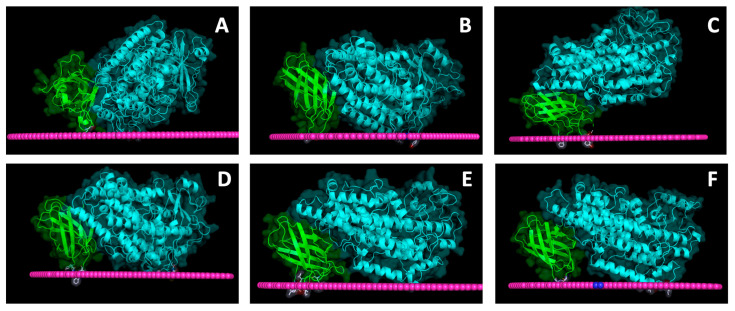
Graphical rendering of soybean 15-LOX (**A**), rabbit 12/15-LOX (**B**), coral 8-LOX (**C**), human 5-LOX (**D**), human 12-LOX (**E**), and human 15-LOX (**F**) bound to a fluid membrane. The three protein structures have been obtained using the PPM 3.0 web server, a tool of Orientations of Proteins in Membranes (OPM) database, https://opm.phar.umich.edu accessed on 25 January 2024 [66,67]. The green- and cyan-colored images correspond to the C- and N-terminal domain of each protein. In pink, the lipid “surface” rendering is displayed. The side chains of the most deeply inserted amino acids into the lipid bilayer (i.e., underneath the membrane plane) are also shown.

**Figure 9 ijms-25-02241-f009:**
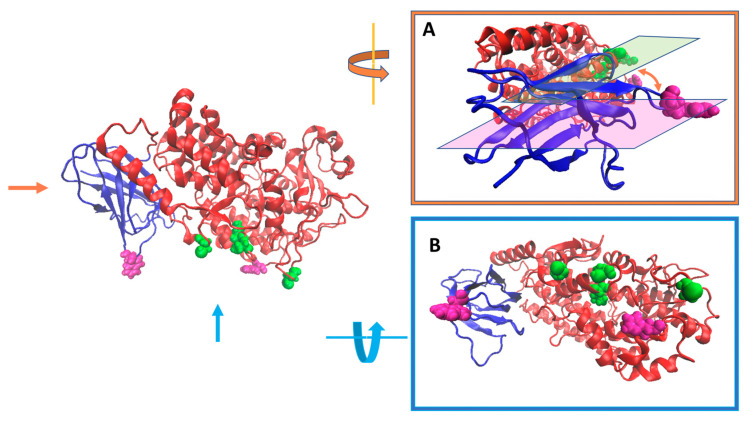
The image on the left represents the human 5-LOX in the same orientation as Figure 2C and Figure 4C. In green the putative cysteines involved in the dimer formation (C159, C416, C418, and C300) are shown. The two residues colored in mauve (W75 and H195) are those supposed to penetrate into the lipid bilayer. In frames (**A**,**B**), the same molecule is shown from the points of view indicated by the orange (**A**) and cyan (**B**) arrows. The angle between the plane perpendicular to the membrane surface (mauve, through W75 and H195) and that of cysteines (green) is highlighted in panel A.

**Table 1 ijms-25-02241-t001:** Membrane binding parameters of different LOX isozymes.

LOX Isozyme	K_d_ (μM)	Membrane *	Condition (T, pH, [Salts], [Ions])	Method **	Reference
Soybean 15-LOX (LOX1)	17.9 ± 2.0	DPPC	Tris HCl pH 8 0.2 M 4 mM CaCl_2_	FRET	Dainese et al. (2010) [72]
Soybean mini-15-LOX	9.2 ± 1.0	DPPC	Tris HCl pH 8 0.2 M 4 mM CaCl_2_	FRET	Dainese et al. (2010) [72]
Soybean mini-15-LOX (Apo form)	45.4 ± 4.3	DPPC	Tris HCl pH 8 0.2 M 4 mM CaCl_2_	FRET	Dainese et al. (2010) [72]
Soybean 15-LOX	0.35 ± 0.03	POPC	Tris HCl pH 7.4 50 mM 0.2 mM CaCl_2_ 22 °C	FRET	Mei et al. (2008) [45]
Soybean 15-LOX	0.36 ± 0.03	DPPC	Tris HCl pH 7.4 50 mM 0.2 mM CaCl_2_ 22 °C	FRET	Mei et al. (2008) [45]
Rabbit 12/15 LOX	0.28 ± 0.02	POPC	Tris HCl pH 7.4 50 mM 0.2 mM CaCl_2_ 22 °C	FRET	Mei et al. (2008) [45]
Rabbit 12/15 LOX	0.35 ± 0.03	DPPC	Tris HCl pH 7.4 50 mM 0.2 mM CaCl_2_ 22 °C	FRET	Mei et al. (2008) [45]
Human 5-LOX	1.15	DPPC	Tris-HCl pH 7.5 50 mM 150 mM NaCl, 0.1 mM EGTA, 0.3 mM CaCl_2_ 22 °C	FRET	Pande et al. (2005) [60]
Human 5-LOX	0.78	POPC	Tris-HCl pH 7.5 50 mM 150 mM NaCl, 0.1 mM EGTA, 0.3 mM CaCl_2_ 22 °C	FRET	Pande et al. (2005) [60]
Human 5-LOX	0.56	PLPC	Tris-HCl pH 7.5 50 mM 150 mM NaCl, 0.1 mM EGTA, 0.3 mM CaCl_2_ 22 °C	FRET	Pande et al. (2005) [60]
Human 5-LOX	0.24	PAPC	Tris-HCl pH 7.5 50 mM 150 mM NaCl, 0.1 mM EGTA, 0.3 mM CaCl_2_ 22 °C	FRET	Pande et al. (2005) [60]
Human 5-LOX	0.32	PDPC	Tris-HCl pH 7.5 50 mM 150 mM NaCl, 0.1 mM EGTA, 0.3 mM CaCl_2_ 22 °C	FRET	Pande et al. (2005) [60]
Human 5-LOX (N-terminal)	≈0.001	PC	Hepes pH 7.4 10 mM 0.1 M NaCl 0.1 mM CaCl_2_ 24 °C	SPR	Kulkarni et al. (2002) [59]
Human 5-LOX (N-terminal)	2.5 ± 0.4	PC	Hepes pH 7.4 10 mM 0.1 M NaCl 0.001 mM CaCl_2_ 24 °C	SPR	Kulkarni et al. (2002) [59]
Coral 8R-LOX	0.28 ± 0.04	PC:PE	Tris-HCl pH 8.0 50 mM	SPR	Rohlik et al. (2023) [62]
Coral 8R-LOX	0.21 ± 0.02	PC:PS	Tris-HCl pH 8.0 50 mM	SPR	Rohlik et al. (2023) [62]
Human 15-LOX-2	0.63 ± 0.02	PC:PS	Tris-HCl pH 8.0 50 mM	SPR	Rohlik et al. (2023) [62]
Coral 8R-LOX	11.4 ± 2.7	POPC:POPS (3:1)	Tris-HCl pH 7.5 50 mM, 500 mM NaCl, 2 mM EDTA 4 mM CaCl2	FRET	Oldham et al. (2005) [25]
Coral 11R-LOX	0.59 ± 0.06	SUV PC	Tris-HCl pH 8.0 50 mM, 100 mM NaCl 25 °C	SPR	Eek et al. (2012) [50]
Coral 11R-LOX	0.12 ± 0.02	SUV PC	Tris-HCl pH 8.0 50 mM, 100 mM NaCl 0.4 mM CaCl_2_ 25 °C	SPR	Eek et al. (2012) [50]

* PC:PE 2:1 (*m*/*m*) phosphatidylcholine:phosphatidylethanolamine; PC:PS 2:1 (*m*/*m*) phosphatidylcholine:phosphatidylserine; DPPC: 1,2- dipalmitoyl-*sn*-glycero-3-phosphocholine; POPC: 1-palmitoy-2-oleyl-*sn*-glycero-3-phosphocholine; PAPC: 1-palmitoyl-2-arachidonoyl-*sn*-glycero-3-phosphocholine; PLPC: 1-palmitoyl-2-linoleoyl-*sn*-glycero-3-phosphocholine; PDPC: 1-palmitoyl-2-docosahexaenoyl-*sn*-glycero-3-phosphocholine; POPS: 1-palmitoyl-2-oleoyl-*sn*-glycero-3-phospho-L-serine; SUV PC Phosphatidylcholine small unilamellar vesicles. ** FRET, fluorescence [Förster] resonance energy transfer; SPR, surface plasmon resonance.

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
