# Peer review of "Conformational Dynamics of Lipoxygenases and Their Interaction with Biological Membranes"

_ijms, 2024, doi:10.3390/ijms25042241_

Round 1
Reviewer 1 Report
Comments and Suggestions for Authors
The article «In Silico study of the conformational dynamics of lipoxygenases and their interaction with biological membranes» is attracted to a relevant topic and has a high applied value.
The aim of this study is to collect new in silico data on a different, much less explored aspect of lipoxygenases (LOXs), that is their interaction with lipid bilayers. To this end, the putative protein-membrane interface of three relevant LOXs (namely human 5-LOX, rabbit 12/15-LOX and soy-bean 15-LOX) with different substrate specificity is discussed. In addition, the main outcomes of new molecular dynamics simulations of the same three LOXs are presented.
The authors described in detail the methodology for molecular dynamics calculations. Although the article does not contain the results of their own in vitro experiments, the authors provide a detailed analysis of literature data, comparing them with their own in silico calculations. The article is well structured, written in sufficient detail and logically, rich in illustrative materials.
For future works, I can express the following wishes to the authors:
1. Inhibitors of the enzymes under study, for example, eicosatetraynoic acid inhibitor, were studied experimentally even by the authors of the article [39], but in the described works on molecular dynamics [for example, 40], their mobility was not studied. Inhibitors can be successfully studied using flexible docking, including cascade docking (when binding in excess of the ligand is modeled, one copy of the inhibitor is planted after another, several in turn). In addition, the authors’ available experimental data on the effects of mutations [41] can be supplemented by docking, since often amino acid substitutions affect the binding of ligands (substrate or inhibitor, as well as fatty acids or glycerol-like compounds).
2. The soybean 15-LOX enzyme globule is larger in size than similar animal proteins. Do authors have any idea what consequences or limitations this might entail?
3. What are the dimensions and likely structure of the short flexible linker? It is clear that, due to its lability, it interferes with crystallization (such examples are known for other proteins of similar topology)
4. Figure 4 shows the binding of the three enzymes to the membrane, indicating the specific residues that penetrate the bilayer. Are there any plans to carry out mutations in the models for these residues - and conduct new series of calculations? How will the binding change, say, with alanine screening, or substitutions with residues with the opposite charge sign?
5. The mention of Reference 58 raises a logical question: has computer modeling of the binding of Boswellic Acid to the enzymes studied been carried out? Docking would be useful in the initial stage, followed by MD. Moreover, some results “is important because it opens new perspectives in the pharmaceutical research of unconventional drugs that could regulate the active site at the distance”
Author Response
Dear Reviewers,
We thank you very much indeed for your helpful suggestions and constructive comments on how to improve the clarity and impact of our manuscript.
In particular, we concur with the suggestion of Referee no. 2 to change our manuscript into a review, by removing our molecular dynamics simulation data. In addition, the manuscript has been now extended to human 12-LOX, human 15-LOX and coral 8-LOX, for which crystallographic data are available.
Finally, as suggested by Referee no. 1, possible future developments that emerged from the available data are discussed.
Point-by-point replies to all criticisms raised are detailed below.
We thank referee no. 1 for his/her positive comments on our work and we have revised the manuscript according to both referees’ suggestions.
In this new version we have transformed the article into a review: our results on the molecular dynamics results are eliminated and instead we have expanded the study on Loxs with previous about human 12-LOX, human 15-LOX and coral 8-LOX. The new title is, therefore, “Conformational dynamics of lipoxygenases and their interaction with biological membranes”
We also agree with the referee about the importance of future in vitro studies on Loxs in order to explore the role of unconventional drugs that could regulate the active site or consider mutations of the residues involved in the membrane binding. At this regard we have included a section entitled “future perspectives” with indications on possible studies to be carried out on these protein systems in light of what has been done so far. We hope that this new version will be appreciated by the referee.
Reviewer 2 Report
Comments and Suggestions for Authors
The manuscript "In Silico study of the conformational dynamics of lipoxygenases and their interaction with biological membranes" by F. Erba et al. presents a nice discussion of lipoxygenases (LOXs) with regard to their interaction with lipid bilayers and substrate specificity. The article presents results from molecular dynamics simulations (MDs) of 3 LOXs (human 5-LOX, rabbit 12/15-LOX and soybean 15-LOX) to infer interactions with lipid bilayers.
In principle the article discusses an interesting topic that could be of interest to the readers of the IJMS. However, the results of the MDs are not relevant and clearly presented with regard to interactions of LOXs with lipid bilayer. In particular, MDs have been performed in explicit solvent, but in the absence of lipid bilayers. The authors present RMSF plots (Fig. 3) to indicate flexible regions of the 3 LOXs under study and the presence of key aromatic residues at flexible loops. One of these (W75) has been verified experimentally as key residue that penetrates into lipid bilayers, whereas 2 other (H53 and F69) are proposed as lipid-interacting residues. So the question is how do the authors support possible interactions with a lipid bilayer without having performed the simulations in the presence of a lipid bilayer?
In the subsequent section, the authors present renderings of the 3 LOXs obtained from the OPM database and the PPM 3.0 web server using the last frame of the 300-ns MD simulations. These models are used to discuss similarities and differences in the angular orientation of the β-sandwich with respect to the plane of the bilayer. So the question here is why the last frame of the simulation is an appropriate structure to model LOX-bilayer interaction? Why not any other random frame sampled through 300 ns, or not the average structure, or a highly populated cluster representative structure? Any answer to these questions however, would be highly speculative, as is the nature of this analysis.
Sampling of LOXs conformational dynamics using MDs in the absence of lipids and use a random selection of frames to calculate protein-lipid interactions is not an appropriate method to infer interactions. Considering that the authors have a deep knowledge of the system, as exhibited by the interesting discussions on experimental evidence published in the literature, I suggest they remove the MD analysis and present a comprehensive review on this topic. In the opposite case, the authors are strongly suggested to carry out MD simulations of the 3 LOXs in the presence of a lipid bilayer and sample their conformational space adequately to monitor protein-lipid interactions thereof. In such case, I would be glad to reconsider reviewing a revised version of the manuscript.
Comments on the Quality of English LanguageThe quality of the article would be greatly improved upon editing from a native English speaker. The manuscript is not difficult to follow but contains several and "peculiar" grammatical issues.
Author Response
Dear Reviewers,
We thank you very much indeed for your helpful suggestions and constructive comments on how to improve the clarity and impact of our manuscript.
In particular, we concur with the suggestion of Referee no. 2 to change our manuscript into a review, by removing our molecular dynamics simulation data. In addition, the manuscript has been now extended to human 12-LOX, human 15-LOX and coral 8-LOX, for which crystallographic data are available.
Finally, as suggested by Referee no. 1, possible future developments that emerged from the available data are discussed.
Point-by-point replies to all criticisms raised are detailed below.
We agree with Referee n.2 about the opportunity to re-modulate our study in a review entitled “Conformational dynamics of lipoxygenases and their interaction with biological membranes”
For this purpose, we have removed our molecular dynamics simulation studies and expanded the discussion of the up-to-date state of the art on this subject including human 12-LOX, human 15-LOX and coral 8-LOX.
Round 2
Reviewer 2 Report
Comments and Suggestions for Authors
The authors made a good choice to revise the manuscript into a very nice review! The article now presents a very comprehensive view of the conformational dynamics of lipoxygenases, including more members than the in original manuscript, and their interactions with membranes that are more clearly shown. The figures are very illustrative and overall, presentation of has been improved significantly. I reckon that this version of the article will be more appealing to the readers of the IJMS, therefore, I can suggest its publication in the present form.
Comments on the Quality of English LanguageLanguage is fine, some minor editing of English is required at some points.